# KNOWLEDGE CONSISTENCY BETWEEN NEURAL NETWORKS AND BEYOND

**Ruofan Liang**[*,a], **Tianlin Li**[*,a], **Longfei Li**[a], **Jing Wang**[b], **and Quanshi Zhang**[a*]
[a]Shanghai Jiao Tong University,  [b]Huawei Technologies

## ABSTRACT

This paper aims to analyze knowledge consistency between pre-trained deep neural networks. We propose a generic definition for knowledge consistency between neural networks at different fuzziness levels. A task-agnostic method is designed to disentangle feature components, which represent the consistent knowledge, from raw intermediate-layer features of each neural network. As a generic tool, our method can be broadly used for different applications. In preliminary experiments, we have used knowledge consistency as a tool to diagnose representations of neural networks. Knowledge consistency provides new insights to explain the success of existing deep-learning techniques, such as knowledge distillation and network compression. More crucially, knowledge consistency can also be used to refine pre-trained networks and boost performance.

## 1 INTRODUCTION

Deep neural networks (DNNs) have shown promise in many tasks of artificial intelligence. However, there is still lack of mathematical tools to diagnose representations in intermediate layers of a DNN, *e.g.* discovering flaws in representations or identifying reliable and unreliable features. Traditional evaluation of DNNs based on the testing accuracy cannot insightfully examine the correctness of representations of a DNN due to leaked data or shifted datasets (Ribeiro et al., 2016).

Thus, in this paper, we propose a method to diagnose representations of intermediate layers of a DNN from the perspective of knowledge consistency. *I.e.* given two DNNs pre-trained for the same task, no matter whether the DNNs have the same or different architectures, we aim to examine whether intermediate layers of the two DNNs encode similar visual concepts.

Here, we define the **knowledge** of an intermediate layer of the DNN as the set of visual concepts that are encoded by features of an intermediate layer. This research focuses on the consistency of "**knowledge**" between two DNNs, instead of comparing the similarity of "**features**." In comparison, the **feature** is referred to as the explicit output of a layer. For example, two DNNs extract totally different features, but these features may be computed using similar sets of visual concepts, *i.e.* encoding consistent knowledge (a toy example of knowledge consistency is shown in the footnote[1]).

Given the same training data, DNNs with different starting conditions usually converge to different knowledge representations, which sometimes leads to the over-fitting problem (Bengio et al., 2014). However, *ideally*, well learned DNNs with high robustness usually comprehensively encode various types of discriminative features and keep a good balance between the completeness and the discrimination power of features (Wolchover, 2017). Thus, the well learned DNNs are supposed to converge to similar knowledge representations.

In general, we can understand knowledge consistency as follows. Let $A$ and $B$ denote two DNNs learned for the same task. $x_A$ and $x_B$ denote two intermediate-layer features of $A$ and $B$, respec-

---

[*]Ruofan Liang and Tianlin Li contribute equally to this research. Quanshi Zhang is the corresponding author with the John Hopcroft Center and MoE Key Lab of Artificial Intelligence AI Institute, Shanghai Jiao Tong University, China. `zqs1022@sjtu.edu.cn`

[1]As a toy example, we show how to revise a pre-trained DNN to generate different features but represent consistent knowledge. The revised DNN shuffles feature elements in a layer $x_{\text{rev}} = Ax$ and shuffles the feature back in the next convolutional layer $\hat{x} = W_{\text{rev}} \cdot x_{\text{rev}}$, where $W_{\text{rev}} = WA^{-1}$; $A$ is a permutation matrix. The knowledge encoded in the shuffled feature is consistent with the knowledge in the original feature.

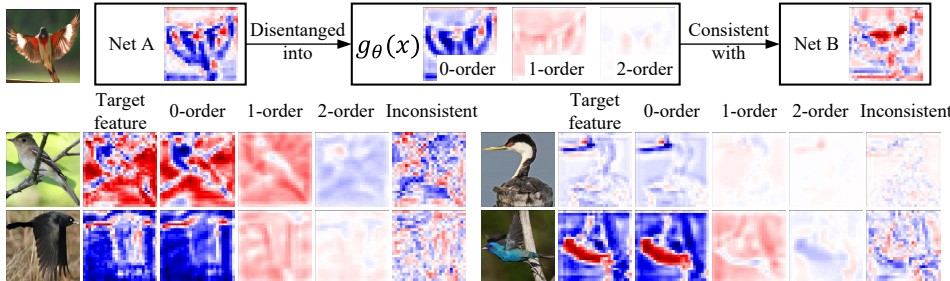

Figure 1: Knowledge consistency. We define, disentangle, and quantify consistent features between two DNNs. Consistent features disentangled from a filter are visualized at different orders (levels) of fuzziness.

tively. Features $x_A$ and $x_B$ can be decomposed as $x_A = \hat{x}_A + \epsilon_A$ and $x_B = \hat{x}_B + \epsilon_B$, where neural activations in feature components $\hat{x}_A$ and $\hat{x}_B$ are triggered by same image regions, thereby representing consistent knowledge. Accordingly, feature components $\hat{x}_A$ and $\hat{x}_B$ are termed **consistent features** between $A$ and $B$. Then, feature components $\epsilon_A$ and $\epsilon_B$ are independent with each other, and they are termed **inconsistent features**. We assume that consistent components $\hat{x}_A, \hat{x}_B$ can reconstruct each other, *i.e.* $\hat{x}_A$ can be reconstructed from $\hat{x}_B$; vice versa.

In terms of applications, knowledge consistency between DNNs can be used to diagnose feature reliability of DNNs. Usually, consistent components $(\hat{x}_A, \hat{x}_B)$ represent common and reliable knowledge. Whereas, inconsistent components $(\epsilon_A, \epsilon_B)$ mainly represent unreliable knowledge or noises.

Therefore, in this paper, we propose a generic definition for knowledge consistency between two pre-trained DNNs, and we develop a method to disentangle consistent feature components from features of intermediate layers in the DNNs. Our method is task-agnostic, *i.e.* (1) our method does not require any annotations *w.r.t.* the task for evaluation; (2) our method can be broadly applied to different DNNs as a supplement evaluation of DNNs besides the testing accuracy. Experimental results supported our assumption, *i.e.* the disentangled consistent feature components are usually more reliable for the task. Thus, our method of disentangling consistent features can be used to boost performance.

Furthermore, to enable a solid research on knowledge consistency, we consider the following issues.

• **Fuzzy consistency at different levels (orders):** As shown in Fig. 1, the knowledge consistency between DNNs needs to be defined at different fuzziness levels, because there is no strict knowledge consistency between two DNNs. The level of fuzziness in knowledge consistency measures the difficulty of transforming features of a DNN to features of another DNN. A low-level fuzziness indicates that a DNN's feature can directly reconstruct another DNN's feature without complex transformations.

• **Disentanglement & quantification:** We need to disentangle and quantify feature components, which correspond to the consistent knowledge at different fuzziness levels, away from the chaotic feature. Similarly, we also disentangle and quantify feature components that are inconsistent.

There does not exist a standard method to quantify the fuzziness of knowledge consistency (*i.e.* the difficulty of feature transformation). For simplification, we use non-linear transformations during feature reconstruction to approximate the fuzziness. To this end, we propose a model $g_k$ for feature reconstruction. The subscript $k$ indicates that $g_k$ contains a total of $k$ cascaded non-linear activation layers. $\hat{x}_A = g_k(x_B)$ represents components, which are disentangled from the feature $x_A$ of the DNN $A$ and can be reconstructed by the DNN $B$'s feature $x_B$. Then, we consider $\hat{x}_A = g_k(x_B)$ to represent consistent knowledge *w.r.t.* the DNN $B$ at the $k$-th fuzziness level (or the $k$-th order). $\hat{x}_A = g_k(x_B)$ is also termed the $k$-order consistent feature of $x_A$ *w.r.t.* $x_B$.

In this way, the most strict consistency is the 0-order consistency, *i.e.* $\hat{x}_A = g_0(x_B)$ can be reconstructed from $x_B$ via a linear transformation. In comparison, some neural activations in the 1-order consistent feature $\hat{x}_A = g_1(x_B)$ are not directly represented by $x_B$ and need to be predicted via a non-linear transformation. The smaller $k$ indicates the less prediction involved in the reconstruction and the stricter consistency. Note that the number of non-linear operations $k$ is just a rough approximation of the difficulty of prediction, since there are no standard methods to quantify prediction difficulties.

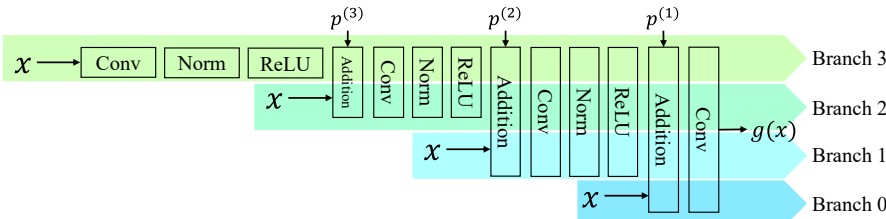

Figure 2: Neural network for disentanglement of consistent features ($K = 3$).

More crucially, we implement the model $g$ as a neural network, where $k$ is set as the number of non-linear layers in $g$. As shown in Fig. 2, $g$ is designed to disentangle and quantify consistent feature components of different orders between DNNs. Our method can be applied to different types of DNNs and explain the essence of various deep-learning techniques.

**1.** Our method provides a new perspective for explaining the effectiveness of knowledge distillation. *I.e.* we explore the essential reason why the born-again network (Furlanello et al., 2018) exhibits superior performance.

**2.** Our method gives insightful analysis of network compression.

**3.** Our method can be used to diagnose and refine knowledge representations of pre-trained DNNs and boost the performance without any additional annotations for supervision.

Contributions of this study can be summarized as follows. (1) In this study, we focus on a new problem, *i.e.* the knowledge consistency between DNNs. (2) We define the knowledge consistency and propose a task-agnostic method to disentangle and quantify consistent features of different orders. (3) Our method can be considered as a mathematical tool to analyze feature reliability of different DNNs. (4) Our method provides a new perspective on explaining existing deep-learning techniques, such as knowledge distillation and network compression.

## 2 RELATED WORK

In spite of the significant discrimination power of DNNs, black-box representations of DNNs have been considered an Achilles' heel for decades. In this section, we will limit our discussion to the literature of explaining or analyzing knowledge representations of DNNs. In general, previous studies can be roughly classified into the following three types.

**Explaining DNNs visually or semantically:** Visualization of DNNs is the most direct way of explaining knowledge hidden inside a DNN, which include gradient-based visualization (Zeiler & Fergus, 2014; Mahendran & Vedaldi, 2015) and inversion-based visualization (Dosovitskiy & Brox, 2016). (Zhou et al., 2015) developed a method to compute the actual image-resolution receptive field of neural activations in a feature map of a convolutional neural network (CNN), which is smaller than the theoretical receptive field based on the filter size. Based on the receptive field, six types of semantics were defined to explain intermediate-layer features of CNNs, including objects, parts, scenes, textures, materials, and colors (Bau et al., 2017; Zhou et al., 2018).

Beyond visualization, some methods diagnose a pre-trained CNN to obtain insight understanding of CNN representations. Fong and Vedaldi (Fong & Vedaldi, 2018) analyzed how multiple filters jointly represented a specific semantic concept. (Selvaraju et al., 2017), (Fong & Vedaldi, 2017), and (Kindermans et al., 2018) estimated image regions that directly contributed the network output. The LIME (Ribeiro et al., 2016) and SHAP (Lundberg & Lee, 2017) assumed a linear relationship between the input and output of a DNN to extract important input units. (Zhang et al., 2019) proposed a number of metrics to evaluate the objectiveness of explanation results of different explanation methods, even when people could not get ground-truth explanations for the DNN.

Unlike previous studies visualizing visual appearance encoded in a DNN or extracting important pixels, our method disentangles and quantifies the consistent components of features between two DNNs. Consistent feature components of different orders can be explicitly visualized.

**Learning explainable deep models:** Compared to the post-hoc explanations of DNNs, some studies directly learn more meaningful CNN representations. Previous studies extracted scene semantics (Zhou et al., 2015) and mined objects (Simon & Rodner, 2015) from intermediate layers. In the capsule net (Sabour et al., 2017), each output dimension of a capsule usually encoded a specific meaning. (Zhang et al., 2018b) proposed to learn CNNs with disentangled intermediate-layer representations. The infoGAN (Chen et al., 2016) and $\beta$-VAE (Higgins et al., 2017) learned interpretable input codes for generative models.

**Mathematical evaluation of the representation capacity:** Formulating and evaluating the representation capacity of DNNs is another emerging direction. (Novak et al., 2018) proposed generic metrics for the sensitivity of network outputs with respect to parameters of neural networks. (Zhang et al., 2017) discussed the relationship between the parameter number and the generalization capacity of deep neural networks. (Arpit et al., 2017) discussed the representation capacity of neural networks, considering real training data and noises. (Yosinski et al., 2014) evaluated the transferability of filters in intermediate layers. Network-attack methods (Koh & Liang, 2017; Szegedy et al., 2014; Koh & Liang, 2017) can also be used to evaluate representation robustness by computing adversarial samples. (Lakkaraju et al., 2017) discovered knowledge blind spots of the knowledge encoded by a DNN via human-computer interaction. (Zhang et al., 2018a) discovered potential, biased representations of a CNN due to the dataset bias. (Deutsch, 2018) learned the manifold of network parameters to diagnose DNNs. (Huang et al., 2019) quantified the representation utility of different layerwise network architectures in the scenario of 3D point cloud processing. Recently, the stiffness (Fort et al., 2019) was proposed to evaluate the generalization of DNNs.

The information-bottleneck theory (Wolchover, 2017; Schwartz-Ziv & Tishby, 2017) provides a generic metric to quantify the information contained in DNNs. The information-bottleneck theory can be extended to evaluate the representation capacity of DNNs (Xu & Raginsky, 2017; Cheng et al., 2018). (Achille & Soatto, 2018) further used the information-bottleneck theory to improve feature representations of a DNN. (Guan et al., 2019; Ma et al., 2019) quantified the word-wise/pixel-wise information discarding during the layerwise feature propagation of a DNN, and used the information discarding as a tool to diagnose feature representations.

The analysis of representation similarity between DNNs is most highly related to our research, which has been investigated in recent years (Montavon et al., 2011). Most previous studies (Wang et al., 2018; Kornblith et al., 2019; Raghu et al., 2017; Morcos et al., 2018) compared representations via linear transformations, which can be considered as 0-order knowledge consistency. In comparison, we focus on high-order knowledge consistency. More crucially, our method can be used to refine network features and explain the success of existing deep-learning techniques.

## 3 ALGORITHM

In this section, we will introduce the network architecture to disentangle feature components of consistent knowledge at a certain fuzziness level, when we use the intermediate-layer feature $x$ of a DNN to reconstruct intermediate-layer feature $x^*$ of another DNN[2]. As shown in Fig. 2, the network $g$ with parameters $\theta$ has a recursive architecture with $K+1$ blocks. The function of the $k$-th block is given as follows.

$$h^{(k)} = W^{(k)} \left[ x + x^{higher} \right], \quad x^{higher} = p^{(k+1)} \text{ReLU} \left( \Sigma_{(k+1)}^{-\frac{1}{2}} h^{(k+1)} \right), \quad k = 0, 1, \ldots, K-1 \quad (1)$$

The output feature is computed using both the raw input and the feature of the higher order $h^{(k+1)}$. $W^{(k)}$ denotes a linear operation without a bias term. The last block is given as $h^{(K)} = W^{(K)}x$. This linear operation can be implemented as either a layer in an MLP or a convolutional layer in a CNN. $\Sigma_{(k+1)} = \text{diag}(\sigma_1^2, \sigma_2^2, \ldots, \sigma_n^2)$ is referred to a diagonal matrix for element-wise variance of $h^{(k+1)}$, where $\sigma_m^2$ the variance of the $m$-th element of $h^{(k+1)}$ through various input images. $\Sigma_{(k+1)}$ is used to normalize the magnitude of neural activations. Because of the normalization, the scalar value $p^{(k+1)}$ roughly controls the activation magnitude of $h^{(k+1)}$ *w.r.t.* $h^{(k)}$. $h^{(0)} = g_\theta(x)$ corresponds to final output of the network.

---

[2]Re-scaling the magnitude of all neural activations does not affect the knowledge encoded in the DNN. Therefore, we normalize $x$ and $x^*$ to zero mean and unit variance to remove the magnitude effect.

In this way, the entire network can be separated into $(K + 1)$ branches (see Fig. 2), where the $k$-th branch ($k \leq K$) contains a total of $k$ non-linear layers. Note that the $k_1$-order consistent knowledge can also be represented by the network with $k_2$-th branch, if $k_1 < k_2$.

In order to disentangle consistent features of different orders, the $k$-th branch is supposed to exclusively encode the $k$-order consistent features without representing lower-order consistent features. Thus, we propose the following loss to guide the learning process.

$$Loss(\theta) = \|g_\theta(x) - x^*\|^2 + \lambda \sum_{k=1}^{K} \left( p^{(k)} \right)^2 \tag{2}$$

where $x$ and $x^*$ denote intermediate-layer features of two pre-trained DNNs. The second term in this loss penalizes neural activations from high-order branches, thereby forcing as much low-order consistent knowledge as possible to be represented by low-order branches.

Furthermore, based on $(K + 1)$ branches of the network, we can disentangle features of $x^*$ into $(K + 2)$ additive components.

$$x^* = g_\theta(x) + x^\Delta, \qquad g_\theta(x) = x^{(0)} + x^{(1)} + \cdots + x^{(K)} \tag{3}$$

where $x^\Delta = x^* - g_\theta(x)$ indicates feature components that cannot be represented by $x$. $x^{(k)}$ denotes feature components that are exclusively represented by the $k$-order branch.

Based on Equation (1), the signal-processing procedure for the $k$-th feature component can be represented as $Conv_k \rightarrow Norm_k \rightarrow ReLU_k \rightarrow p^{(k)} \rightarrow Conv_{k-1} \rightarrow \cdots \rightarrow ReLU_1 \rightarrow p^{(1)} \rightarrow Conv_0$ (see Fig. 2). Therefore, we can disentangle from Equation (1) all linear and non-linear operations along the entire $k$-th branch as follows, and in this way, $x^{(k)}$ can be considered as the $k$-order consistent feature component.

$$x^{(k)} = W^{(0)} \left( \prod_{k'=1}^{k} p^{(k')} A^{(k')} \Sigma_{(k')}^{-\frac{1}{2}} W^{(k')} \right) x \tag{4}$$

where $A^{(k')} = \mathrm{diag}(a_1, a_2, \ldots, a_M)$ is a diagonal matrix. Each diagonal element $a_m \in \{0, 1\}$ represents the binary information-passing state of $x_m$ through an ReLU layer ($1 \leq m \leq M$). Each element is given as $a_m = \mathbf{1}(v_m > 0)$, where $v = \Sigma_{(k')}^{-\frac{1}{2}} h^{(k')}$. Note that $ReLU(W^k(x + x^{higher})) \neq ReLU(W^k(x)) + ReLU(W^k(x^{higher}))$, but $A^{(k')}(W^k(x + x^{higher})) = A^{(k')}W^k x + A^{(k')}W^k x^{higher}$. Thus, we record such information-passing states to decompose feature components.

## 4 COMPARATIVE STUDIES

As a generic tool, knowledge consistency based on $g$ can be used for different applications. In order to demonstrate utilities of knowledge consistency, we designed various comparative studies, including (1) diagnosing and debugging pre-trained DNNs, (2) evaluating the instability of learning DNNs, (3) feature refinement of DNNs, (4) analyzing information discarding during the compression of DNNs, and (5) explaining effects of knowledge distillation in knowledge representations.

A total of five typical DNNs for image classification were used, *i.e.* the AlexNet (Krizhevsky et al., 2012), the VGG-16 (Simonyan & Zisserman, 2015), and the ResNet-18, ResNet-34, ResNet-50 (He et al., 2016). These DNNs were learned using three benchmark datasets, which included the CUB200-2011 dataset (Wah et al., 2011), the Stanford Dogs dataset (Khosla et al., 2011), and the Pascal VOC 2012 dataset (Everingham et al., 2015). Note that both training images and testing images were cropped using bounding boxes of objects. We set $\lambda = 0.1$ for all experiments, except for feature reconstruction of AlexNet (we set $\lambda = 8.0$ for AlexNet features). It was because the shallow model of the AlexNet usually had significant noisy features, which caused considerable inconsistent components.

### 4.1 NETWORK DIAGNOSIS BASED ON KNOWLEDGE CONSISTENCY

The most direct application of knowledge consistency is to use a strong (well learned) DNN to diagnose representation flaws hidden in a weak DNN. This is of special values in real applications, *e.g.* shallow (usually weak) DNNs are more suitable to be adopted to mobile devices than deep DNNs.

| (a) Blind spots of the weak DNN | (a) Unreliable/noisy features of the weak DNN |

Figure 3: Unreliable components and blind spots of a weak DNN (AlexNet) *w.r.t.* a strong DNN (ResNet-34). Please see Section 4.1 for definitions of "unreliable components" and "blind spots." (left) When we used features of the weak DNN to reconstruct features of the strong DNN, we visualize raw features of the strong DNN, feature components that can be reconstructed, and blind spots ($x^{\Delta}$) disentangled from features of the strong DNN. The weak DNN mainly encoded the head appearance and ignored others. We can find that some patterns in the torso are blind spots of the weak DNN. (right) When we used features of the strong DNN to reconstruct features of the weak DNN, we visualize raw features of the weak DNN, feature components that can be reconstructed, and unreliable features ($x^{\Delta}$) disentangled from features of the weak DNN. Based on visualization results, unreliable features of the weak DNN usually repesent noisy activations.

Let two DNNs be trained for the same task, and one DNN significantly outperforms the other DNN. We assume that the strong DNN has encoded ideal knowledge representations of the target task. The weak DNN may have the following two types of representation flaws.

• **Unreliable features** in the weak DNN are defined as feature components, which cannot be reconstructed by features of the strong DNN. (see Appendix B for detailed discussions).

• **Blind spots** of the weak DNN are defined as feature components in the strong DNN, which are inconsistent with features of the weak DNN. These feature components usually reflect blind spots of the knowledge of the weak DNN (see Appendix B for detailed discussions).

For implementation, we trained DNNs for fine-grained classification using the CUB200-2011 dataset (Wah et al., 2011) (without data augmentation). We considered the AlexNet (Krizhevsky et al., 2012) as the weak DNN (56.97% top-1 accuracy), and took the ResNet-34 (He et al., 2016) as the strong DNN (73.09% top-1 accuracy).

Please see Fig. 3. We diagnosed the output feature of the last convolutional layer in the AlexNet, which is termed $x_A$. Accordingly, we selected the last $14 \times 14 \times 256$ feature map of the ResNet-34 (denoted by $x_B$) for the computation of knowledge consistency, because $x_A$ and $x_B$ had similar map sizes. We disentangled and visualized unreliable components from $x_A$ (*i.e.* inconsistent components in $x_A$). We also visualized components disentangled from $x_B$ (*i.e.* inconsistent components in $x_B$), which corresponded to blind spots of the weak DNN's knowledge.

Furthermore, we conducted two experiments to further verify the claim of blind spots and unreliable features in a DNN. The first experiment aimed to verify blind spots. The basic idea of this experiment was to examine the increase of the classification accuracy, when we added information of blind spots to the raw feature. We followed above experimental settings, which used the intermediate-layer feature $x_A$ of the AlexNet (with 56.97% top-1 accuracy) to reconstruct the intermediate-layer feature $x_B$ of Resnet-34 (with 73.09% top-1 accuracy). Then, inconsistent feature components were termed blind spots. In this way, we added feature components of blind spots back to the AlexNet's feature (adding these features back to $g_\theta(x_A)$), and then learned a classifier upon the new feature. To enable a fair comparison, the classifier had the same architecture as the AlexNet's modules above $x_A$, and during the learning process, we fixed parameters in DNN A and $\theta$ to avoid the revision of such parameters affecting the performance. We found that compared to the raw feature $x_A$ of the AlexNet, the new feature boosted the classification accuracy by 16.1%.

The second experiment was conducted to verify unreliable features. The basic idea of this experiment was to examine the increase of the classification accuracy, when we removed information of unreliable features from the raw feature. We designed two classifiers, which had the same architecture as the AlexNet's modules above $x_A$. We fed the raw feature $x_A$ to the first classifier. Then, we removed unreliable feature components from $x_A$ (*i.e.* obtaining $g_\theta(x_B)$), and fed the revised feature to the second classifier. We learned these two classifiers, and classifiers with and without unreliable features exhibited classification accuracy of 60.3% and 65.6%, respectively.

Therefore, above two experiments successfully demonstrated that both the insertion of blind spots and the removal of unreliable features boosted the classification accuracy.

| Learning DNNs from different initializations | | | | |
|---|---|---|---|---|
| conv4 @ AlexNet | conv5 @ AlexNet | conv4-3 @ VGG-16 | conv5-3 @ VGG-16 | last conv @ ResNet-34 |
| 0.086 | 0.116 | 0.124 | 0.196 | 0.776 |
| Learning DNNs using different training data | | | | |
| conv4 @ AlexNet | conv5 @ AlexNet | conv4-3 @ VGG-16 | conv5-3 @ VGG-16 | last conv @ ResNet-34 |
| 0.089 | 0.155 | 0.121 | 0.198 | 0.275 |

Table 1: Instability of learning DNNs from different initializations and instability of learning DNNs using different training data. Without a huge training set, networks with more layers (*e.g.* ResNet-34) usually suffered more from the over-fitting problem.

| | conv4 @ AlexNet | conv5 @ AlexNet | conv4-3 @ VGG-16 | conv5-3 @ VGG-16 | last conv @ ResNet-34 |
|---|---|---|---|---|---|
| $Var(x^{(0)})$ | 105.80 | 424.67 | 1.06 | 0.88 | 0.66 |
| $Var(x^{(1)})$ | 10.51 | 73.73 | 0.07 | 0.03 | 0.10 |
| $Var(x^{(2)})$ | 1.92 | 43.69 | 0.02 | 0.004 | 0.03 |
| $Var(x^{\Delta})$ | 11.14 | 71.37 | 0.16 | 0.22 | 2.75 |

Table 2: Magnitudes of consistent features of different orders, when we train DNNs from different initializations. Features in different layers have significantly different variance magnitudes. Most neural activations of the feature belong to low-order consistent components.

## 4.2 STABILITY OF LEARNING

The stability of learning DNNs is of considerable values in deep learning, *i.e.* examining whether or not all DNNs represent the same knowledge, when people repeatedly learn multiple DNNs for the same task. High knowledge consistency between DNNs usually indicates high learning stability.

More specifically, we trained two DNNs $(A, B)$ with the same architecture for the same task. Then, we disentangled inconsistent feature components $x_A^{\Delta} = x_A - g_{\theta}(x_B)$ and $x_B^{\Delta} = x_B - g_{\theta}(x_A)$ from their features $x_A$ and $x_B$ of a specific layer, respectively. Accordingly, $g_{\theta}(x_B)$ and $g_{\theta}(x_A)$ corresponded to consistent feature components in $x_A$ and $x_B$, respectively, whose knowledge was shared by two networks. The inconsistent feature $x_A^{\Delta}$ was quantified by the variance of feature element through different units of $x_A^{\Delta}$ and through different input images $Var(x_A^{\Delta}) \stackrel{\text{def}}{=} \mathbb{E}_{I,i}[(x_{A,I,i}^{\Delta} - \mathbb{E}_{I',i'}[x_{A,I',i'}^{\Delta}])^2]$, where $x_{A,I,i}^{\Delta}$ denotes the $i$-th element of $x_A^{\Delta}$ given the image $I$. We can use $Var(x_A^{\Delta})/Var(x_A)$ to measure the instability of learning DNNs $A$ and $B$.

In experiments, we evaluated the instability of learning the AlexNet (Krizhevsky et al., 2012), the VGG-16 (Simonyan & Zisserman, 2015), and the ResNet-34 (He et al., 2016). We considered the following cases.

**Case 1,** learning DNNs from different initializations using the same training data: For each network architecture, we trained multiple networks using the CUB200-2011 dataset (Wah et al., 2011). The instability of learning DNNs was reported as the average of instability over all pairs of neural networks.

**Case 2,** learning DNNs using different sets of training data: We randomly divided all training samples in the CUB200-2011 dataset (Wah et al., 2011) into two subsets, each containing 50% samples. For each network architecture, we trained two DNNs $(A, B)$ for fine-grained classification (without pre-training). The instability of learning DNNs was reported as $[Var(x_A^{\Delta})/Var(x_A) + Var(x_B^{\Delta})/Var(x_B)]/2$.

Table 1 compares the instability of learning different DNNs. Table 2 reports the variance of consistent components of different orders. We found that the learning of shallow layers in DNNs was usually more stable than the learning of deep layers. The reason may be as follows. A DNN with more layers usually can represent more complex visual patterns, thereby needing more training samples. Without a huge training set (*e.g.* the ImageNet dataset (Krizhevsky et al., 2012)), a deep network may be more likely to suffer from the over-fitting problem, which causes high variances in Table 1, *i.e.* DNNs with different initial parameters may learn different knowledge representations.

## 4.3 FEATURE REFINEMENT BASED ON KNOWLEDGE CONSISTENCY

Knowledge consistency can also be used to refine intermediate-layer features of pre-trained DNNs. Given multiple DNNs pre-trained for the same task, feature components, which are consistent with various DNNs, usually represent common knowledge and are reliable. Whereas, inconsistent feature components *w.r.t.* other DNNs usually correspond to unreliable knowledge or noises. In this way,

| | VGG-16 conv4-3 | VGG-16 conv5-2 | ResNet-18 | ResNet-34 | ResNet-50 |
|---|---|---|---|---|---|
| Network $A$ | 43.15 | | 34.74 | 31.05 | 29.98 |
| Network $B$ | 42.89 | | 35.00 | 30.46 | 31.15 |
| $x^{(0)}$ | **45.15** | **44.48** | 38.16 | 31.49 | 30.40 |
| $x^{(0)} + x^{(1)}$ | 44.98 | 44.22 | **38.45** | 31.76 | 31.77 |
| $x^{(0)} + x^{(1)} + x^{(2)}$ | 45.06 | 44.32 | 38.23 | **31.96** | **31.84** |

Table 3: Classification accuracy by using the original and the refined features. Features of DNN $A$ were used to reconstruct features of DNN $B$. For residual networks, we selected the last feature map with a size of $14 \times 14$ as the target for refinement. All DNNs were learned without data augmentation or pre-training. The removal of effects of additional parameters in $g_\theta$ is discussed in Section 4.3– *fairness of comparisons* and Appendix A. Consistent components slightly boosted the performance.

intermediate-layer features can be refined by removing inconsistent components and exclusively using consistent components to accomplish the task.

More specifically, given two pre-trained DNNs, we use the feature of a certain layer in the first DNN to reconstruct the corresponding feature of the second DNN. The reconstructed feature is given as $\hat{x}$. In this way, we can replace the feature of the second DNN with the reconstructed feature $\hat{x}$, and then use $\hat{x}$ to simultaneously learn all subsequent layers in the second DNN to boost performance. Note that for clear and rigorous comparisons, we only disentangled consistent feature components from the feature of a single layer and refined the feature. It was because the simultaneous refinement of features of multiple layers would increase it difficult to clarify the refinement of which feature made the major contribution.

In experiments, we trained DNNs with various architectures for image classification, including the VGG-16 (Simonyan & Zisserman, 2015), the ResNet-18, the ResNet-34, and the ResNet-50 (He et al., 2016). We conducted the following two experiments, in which we used knowledge consistency to refine DNN features.

**Exp. 1, removing unreliable features (noises):** For each specific network architecture, we trained two DNNs using the CUB200-2011 dataset (Wah et al., 2011) with different parameter initializations. Consistent components were disentangled from the original feature of a DNN and then used for image classification. As discussed in Section 4.1, consistent components can be considered as refined features without noises. We used the refined features as input to finetune the pre-trained upper layers in the DNN $B$ for classification. Table 3 reports the increase of the classification accuracy by using the refined feature. The refined features slightly boosted the performance.

**Fairness of comparisons:** (1) To enable fair comparisons, we first trained $g_\theta$ and then kept $g_\theta$ **unchanged** during the finetuning of classifiers, thereby **fixing** the refined features. Otherwise, allowing $g_\theta$ to change would be equivalent to adding more layers/parameters to DNN $B$ for classification. We needed to eliminate such effects for fair comparisons, which avoided the network from benefitting from **additional layers/parameters**. (2) As baselines, we also further finetuned the corresponding upper layers of DNNs $A$ and $B$ to evaluate their performance. Please see Appendix A for more discussions about how we ensured the fairness.

**Exp. 2, removing redundant features from pre-trained DNNs:** A typical deep-learning methodology is to finetune a DNN for a specific task, where the DNN is pre-trained for multiple tasks (including both the target and other tasks). This is quite common in deep learning, *e.g.* DNNs pre-trained using the ImageNet dataset (Deng et al., 2009) are usually finetuned for various tasks. However, in this case, feature components pre-trained for other tasks are redundant for the target task and will affect the further finetuning process.

Therefore, we conducted three new experiments, in which our method removed redundant features *w.r.t.* the target task from the pre-trained DNN. In Experiment 2.1 (namely *VOC-animal*), let two DNNs $A$ and $B$ be learned to classify 20 object classes in the Pascal VOC 2012 dataset (Everingham et al., 2015). We were also given object images of six animal categories (*bird, cat, cow, dog, horse, sheep*). We fixed parameters in DNNs $A$ and $B$, and our goal was to use fixed DNNs $A$ and $B$ to generate clean features for animal categories without redundant features.

Let $x_A$ and $x_B$ denote intermediate-layer features of DNNs $A$ and $B$. Our method used $x_A$ to reconstruct $x_B$. Then, the reconstructed result $g_\theta(x_A)$ corresponded to reliable features for animals, while inconsistent components $x^\Delta$ indicated features of other categories. We used clean features $g_\theta(x_A)$ to learn the animal classifier. $g_\theta$ were fixed during the learning of the animal classifier to

|                                  | VGG-16 conv4-3 | | | VGG-16 conv5-2 | | |
|                                  | VOC-animal | Mix-CUB | Mix-Dogs | VOC-animal | Mix-CUB | Mix-Dogs |
|----------------------------------|------------|---------|----------|------------|---------|----------|
| Features from the network $A$    | 51.55      | 44.44   | 15.15    | 51.55      | 44.44   | 15.15    |
| Features from the network $B$    | 50.80      | 45.93   | 15.19    | 50.80      | 45.93   | 15.19    |
| $x^{(0)} + x^{(1)} + x^{(2)}$    | **59.38**  | **47.50** | **16.53** | **60.18**  | **46.65** | **16.70** |
|                                  | ResNet-18  | | | ResNet-34 | | |
|                                  | VOC-animal | Mix-CUB | Mix-Dogs | VOC-animal | Mix-CUB | Mix-Dogs |
| Features from the network $A$    | 37.65      | 31.93   | 14.20    | 39.42      | 30.91   | 12.96    |
| Features from the network $B$    | 37.22      | 32.02   | 14.28    | 35.95      | 27.74   | 12.46    |
| $x^{(0)} + x^{(1)} + x^{(2)}$    | **53.52**  | **38.02** | **16.17** | **49.98**  | **33.98** | **14.21** |

Table 4: Top-1 classification accuracy before and after removing redundant features from DNNs. Original DNNs were learned from scratch without data augmentation. For residual networks, we selected the last feature map with a size of $14 \times 14$ as the target for feature refinement. The removal of effects of additional parameters in $g_\theta$ is discussed in Section 4.3–*fairness of comparisons* and Appendix A. Consistent features significantly alleviated the over-fitting problem, thereby exhibiting superior performance.

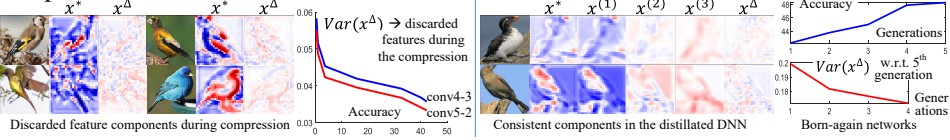

Figure 4: Effects of network compression and knowledge distillation. (left) We visualized the discarded feature components when 93.3% parameters of the DNN were eliminated. We found that the discarded components looked like trivial features and noises. In the curve figure, a low value of $Var(x^\Delta)$ indicates a lower information discarding during the compression, thereby exhibiting a higher accuracy. (right) We visualized and quantified knowledge blind spots of born-again networks in different generations. Nets in new generations usually had fewer blind spots than old nets.

enable a fair comparison. In comparison, the baseline method directly used either $x_A$ or $x_B$ to finetune the pre-trained DNN to classify the six animal categories.

In Experiment 2.2 (termed *Mix-CUB*), two original DNNs were learned using both the CUB200-2011 dataset (Wah et al., 2011) and the Stanford Dogs dataset (Khosla et al., 2011) to classify both 200 bird species and 120 dog species. Then, our method used bird images to disentangle feature components for birds and then learned a new fine-grained classifier for birds. The baseline method was implemented following the same setting as in *VOC-animal*. Experiment 2.3 (namely *Mix-Dogs*) was similar to *Mix-CUB*. Our method disentangled dog features away from bird features to learn a new dog classifier. In all above experiments, original DNNs were learned from scratch without data augmentation. Table 4 compares the classification accuracy of different methods. It shows that our method significantly alleviated the over-fitting problem and outperformed the baseline.

## 4.4 ANALYZING INFORMATION DISCARDING OF NETWORK COMPRESSION

Network compression is an emerging research direction in recent years. Knowledge consistency between the compressed network and the original network can evaluate the discarding of knowledge during the compression process. *I.e.* people may visualize or analyze feature components in the original network, which are not consistent with features in the compressed network, to represent the discarded knowledge in the compressed network.

In experiments, we trained the VGG-16 using the CUB200-2011 dataset (Wah et al., 2011) for fine-grained classification. Then, we compressed the VGG-16 using the method of (Han et al., 2016) with different pruning thresholds. We used features of the compressed DNN to reconstruct features of the original DNN. Then, inconsistent components disentangled from the original DNN usually corresponded to the knowledge discarding during the compression process. Fig. 4(left) visualizes the discarded feature components. We used $Var(x^\Delta)$ (defined in Section 4.2) to quantify the information discarding. Fig. 4 compares the decrease of accuracy with the discarding of feature information.

## 4.5 EXPLAINING KNOWLEDGE DISTILLATION VIA KNOWLEDGE CONSISTENCY

As a generic tool, our method can also explain the success of knowledge distillation. In particular, (Furlanello et al., 2018) proposed a method to gradually refine a neural network via recursive knowledge distillation. *I.e.* this method recursively distills the knowledge of the current net to a

Figure 5: Consistent and inconsistent feature components disentangled between DNN B learned for binary classification and DNN A learned for fine-grained classification.

new net with the same architecture and distilling the new net to an even newer net. The new(er) net is termed a *born-again neural network* and learned using both the task loss and the distillation loss. Surprisingly, such a recursive distillation process can substantially boost the performance of the neural network in various experiments.

In general, the net in a new generation both inherits knowledge from the old net and learns new knowledge from the data. The success of the born-again neural network can be explained as that knowledge representations of networks are gradually enriched during the recursive distillation process. To verify this assertion, in experiments, we trained the VGG-16 using the CUB200-2011 dataset (Wah et al., 2011) for fine-grained classification. We trained born-again neural networks of another four generations[3]. We disentangled feature components in the newest DNN, which were not consistent with an intermediate DNN. Inconsistent components were considered as blind spots of knowledge representations of the intermediate DNN and were quantified by $Var(x^\Delta)$. Fig. 4(right) shows $Var(x^\Delta)$ of DNNs in the 1st, 2nd, 3rd, and 4th generations. Inconsistent components were gradually reduced after several generations.

### 4.6 CONSISTENT AND INCONSISTENT FEATURES BETWEEN DIFFERENT TASKS

In order to visualize consistent and inconsistent features between different tasks (*i.e.* the fine-grained classification and the binary classification), we trained DNN A (a VGG-16 network) to simultaneously classify 320 species, including 200 bird species in the CUB200-2011 dataset (Wah et al., 2011) and 120 dog species in the Stanford Dogs dataset (Khosla et al., 2011), in a fine-grained manner. On the other hand, we learned DNN B (another VGG-16 network) for the binary classification of the bird and the dog based on these two datasets. We visualized the feature maps of consistent and inconsistent feature components between the two DNNs in Fig. 5. Obviously, DNN A encoded more knowledge than DNN B, thereby $x_A$ reconstructing more features than $x_B$.

## 5 CONCLUSION

In this paper, we have proposed a generic definition of knowledge consistency between intermediate-layers of two DNNs. A task-agnostic method is developed to disentangle and quantify consistent features of different orders from intermediate-layer features. Consistent feature components are usually more reliable than inconsistent components for the task, so our method can be used to further refine the pre-trained DNN without a need for additional supervision. As a mathematical tool, knowledge consistency can also help explain existing deep-learning techniques, and experiments have demonstrated the effectiveness of our method.

ACKNOWLEDGEMENTS

This work was partially supported by National Natural Science Foundation of China (U19B2043 and 61906120) and Huawei Technologies.

---

[3]Because (Furlanello et al., 2018) did not clarify the distillation loss, we applied the distillation loss in (Hinton et al., 2014) following parameter settings $\tau = 1.0$ in (Mishra & Marr, 2018), *i.e.* $Loss = Loss^{\text{classify}} + 0.5Loss^{\text{distill}}$.

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

## A    FAIRNESS OF EXPERIMENTS IN SECTION 4.3

In Section 4.3, we used knowledge consistency to refine intermediate-layer features of pre-trained DNNs. Given multiple DNNs pre-trained for the same task, feature components, which are consistent with various DNNs, usually represent common knowledge and are reliable. In this way, intermediate-layer features can be refined by removing inconsistent components and exclusively using consistent components to accomplish the task.

Both Experiment 1 and Experiment 2 followed the same procedure. *I.e.* we trained two DNNs $A$ and $B$ for the same task. Then, we extracted consistent feature components $g(x_A)$ when we used the DNN $A$'s feature to reconstruct the DNN $B$'s feature. We compared the classification performance of the DNN $A$, the DNN $B$, and the classifier learned based on consistent features $g(x_A)$.

However, an issues of fairness may be raised, *i.e.* when we added the network $g$ upon the DNN $A$, this operation increased the depth of the network. Thus, the comparison between the revised DNN and the original DNN $A$ may be unfair.

In order to ensure a fair comparison, we applied following experimental settings.
• For the evaluation of DNNs $A$ and $B$, both the DNNs were further refined using target training samples in Experiments 1 and 2. • For the evaluation of our method, without using any annotations of the target task, we first trained $g(x_A)$ to use $x_A$ to reconstruct $x_B$ and disentangle consistent feature components. Then, we fixed all parameters of DNNs $A$, $B$, and $g$, and only used output features of $g(x_A)$ to train a classifier, in order to test the discrimination power of output features of $g(x_A)$.

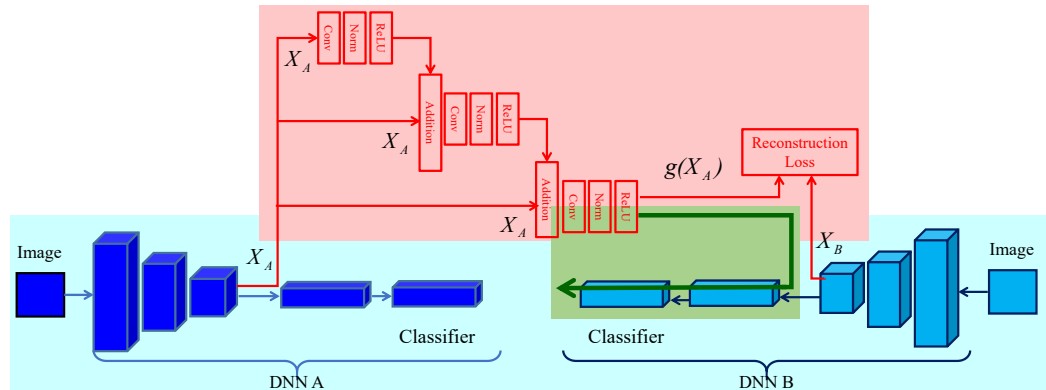

As shown in the above figure, for the evaluation of our method, first, we were not given any training annotations for the target task. Without any prior knowledge of the target task, we used the pre-trained DNNs (blue area in the above figure) to train the network $g$ (red area in the above figure). Then, we fixed parameters of DNNs in both red area and blue area in the above figure. Finally, we received training samples of the target task, and used them to exclusively train the classifier (the blue area in the above figure).

In short, the learning of the network $g$ was independent with the target task, and parameters of $A$, $B$, and $g$ were not refine-tuned during the learning of the new classifier.

Therefore, comparisons in Tables 3 and 4 fairly reflects the discrimination power of raw features of DNNs $A$ and $B$ and consistent features produced by $g(x_A)$.

## B    BLIND SPOTS AND UNRELIABLE FEATURES

By our assumption, strong DNNs can encode true knowledge, while there are more representation flaws in weak DNNs. If we try to use the intermediate features of a weak DNN to reconstruct intermediate features of a strong DNN, the strong DNN features cannot be perfectly reconstructed due to the inherent *blind spots* of the weak DNN. On the other hand, when we use features of a strong DNN to reconstruct a weak DNN's features, the unreconstructed feature components in these

reconstructed features also exist. These unreconstructed parts are not modeled in the strong DNN, and they are termed *unreliable features* by us.

