# OpenReview forum: "Knowledge Consistency between Neural Networks and Beyond"
_ICLR.cc/2020/Conference — Accept (Poster)_

### Official Review · AnonReviewer2 · 2019-10-24
**Official Blind Review #2**

**Rating:** 6

**Review:**

The goal of this paper is to analyze knowledge consistency between pretrained deep neural nets. In order to do so the paper trains neural networks to predict a hidden layer of one DNN using a hidden layer of another DNN. The model is interesting in that it is multi layer but it also allows decomposing its prediction as the sum of outputs of neural nets with different numbers of hidden layers. The prediction is dubbed "consistent features", which are further decomposed in consistent features of different complexity levels, while the error is dubbed "inconsistent features".

The paper then attempts to use this decomposition in "consistent" and "inconsistent" features in a number of ways. Assuming hidden layers of DNN A are being used to predict layer of DNN B, if A is stronger than B then the inconsistent features of B are claimed to be "unreliable features". If A is weaker, then the inconsistent features of B are claimed to be "blind spots". A figure is given to support this intuition, but I think some real evidence would have to be collected to support a claim like this.

Similarly the authors analyze model compression and distillation with their technique. They are able to show a decrease in the variance of inconsistent features as a models get progressively compressed or as a models get better during generations of distillation. Again the evidence that of the usefulness of this analysis seems very limited. The visualizations of consistent and inconsistent components don't seem to give any clear evidence.

The most interesting part of the paper shows experiments from boosting performance of a model by finetuning on features consistent with another model. None of the improvements seem close to state of the art though, and might just be some byproduct of model ensembling.

Overall the paper is very interesting, but more convincing experimental results would have to be collected to prove that the method is actually useful.

**Experience Assessment:**

I have published one or two papers in this area.

**Review Assessment: Checking Correctness Of Derivations And Theory:**

I carefully checked the derivations and theory.

**Review Assessment: Checking Correctness Of Experiments:**

I assessed the sensibility of the experiments.

**Review Assessment: Thoroughness In Paper Reading:**

I read the paper thoroughly.

---

> ### Author Response · Authors · 2019-11-12
> **Thanks for your great efforts on the review of this paper. We are glad to answer all your concerns one by one.**
>
> Thanks for your great efforts on the review of this paper. We are glad to answer all your concerns one by one.
>
> Q: About further explanation for blind spots and unreliable features in a DNN. “The paper then attempts to use this decomposition in consistent and inconsistent features in a number of ways. Assuming hidden layers of DNN A are being used to predict layer of DNN B, if A is stronger than B then the inconsistent features of B are claimed to be unreliable features. If A is weaker, then the inconsistent features of B are claimed to be blind spots. A figure is given to support this intuition, but I think some real evidence would have to be collected to support a claim like this.”
>
> A: This is a good question. We have followed your suggestions to conduct two additional experiments to further verify the claim of “blind spots” and “unreliable features” in a DNN, although we have visualized feature maps corresponding to “blind spots” and “unreliable features” in Figure 3.
>
> Experiment 1 for the verification of blind spots: The basic idea of this experiment was to examine the increase of the classification accuracy, when we added information of blind spots to the raw feature. We followed the settings mentioned in sec. 4.1. We used the intermediate-layer feature of a weak DNN A to reconstruct the intermediate-layer feature of a strong DNN B via our reconstruction model g. Then, inconsistent feature components of B were termed "blind spots." In this way, we added "blind spots" feature components back to DNN A’s feature (adding these features back to g(X_A)), and then learned a classifier upon the new feature. For fair comparisons, the classifier had the same architecture as DNN A’s modules above X_A; during the learning process, we fixed parameters in DNN A and g to avoid the revision of such parameters affecting the performance. We found that compared to the raw feature X_A of DNN A, the new feature boosted the classification accuracy by 16.1%.
>
> Experiment 2 for the verification of unreliable features: The basic idea of this experiment was to examine the increase of the classification accuracy, when we removed information of unreliable features from the raw feature. We designed two classifiers, which had the same architecture as DNN A’s modules above X_A. We fed the raw feature X_A of DNN A to the first classifier. Then, we removed unreliable feature components from X_A (i.e., obtaining g(X_B)), and fed the revised feature to the second classifier. We learned these two classifiers. Classifiers with and without unreliable features exhibited the classification accuracy of 60.3% and 65.6%, respectively.
>
> Therefore, above two experiments successfully demonstrated that both the insertion of blind spots and the removal of unreliable features boosted the classification accuracy. We have introduced the new experiments in Section 4.1.
>
>
> Q: “Similarly, the authors analyze model compression and distillation with their technique …  Again the evidence that of the usefulness of this analysis seems very limited. The visualizations of consistent and inconsistent components don't seem to give any clear evidence.”
>
> A: Thanks for your careful review, but it seems that this concern may be raised by the misunderstanding of the motivation of this research. As a generic tool, our method revealed the connection between knowledge consistency and model accuracy. As models were progressively compressed and as models got better during generations of distillation, inconsistent features indeed decreased, which verified our hypothesis. Thus, our method provides a new perspective to understand network compression and knowledge distillation.
>
>
> Q: About the performance of object classification. “The most interesting part of the paper shows experiments from boosting performance of a model by finetuning on features consistent with another model. None of the improvements seem close to state of the art though, and might just be some byproduct of model ensembling.”
>
> A: This concern may be raised by the misunderstanding of the motivation of this research. This research aims to propose a simple yet general method to analyze and explain the knowledge consistency, instead of designing a new model/method to boost the classification accuracy. To reduce the effect of other unrelated factors, we just used classic models and datasets without either abnormal architectures or any tricks (e.g., pre-training and data augmentation). In this way, we did not make a comparison with state-of-the-art results for classification.
>
> As for ensembling, our method can be treated as an ensembling method, to some extent. However, our motivation is to disentangle and qualify knowledge consistency, which is the distinct contribution of this study. In addition, our method also does other interesting things, such as 1) diagnosing and refining pre-trained DNNs without any additional annotations, 2) the analysis of network compression and knowledge distillation.

---

### Official Review · AnonReviewer3 · 2019-10-25
**Official Blind Review #3**

**Rating:** 8

**Review:**

The submission proposes a method for extracting "knowledge consistency" in neural networks and using that toward analyzing different aspects of them, eg understanding the representations, explaining knowledge distillation, and analyzing network compression. What's defined as consistent knowledge is essentially the stable parts of representations of different networks trained for the same task (stable-->consistent). I found the submission insightful, well executed, and backed up by many experimental results.

Strengths:
A) the proposed concept, extracting the *consistency* among representation of different networks, is interested and using that towards understanding what's going on under-the-hood of neural networks makes sense. I'm not aware of a similar existing method and didn't see one among the citations, so my current assumption is that this proposal is pretty novel.

B) the extracted representational consistency quantity appears to be meaningful and strong, as authors were able to use it toward explaining several existing phenomena (e.g. knowledge distillation, network compression, etc).

C) Authors perform extensive experimental studies on various aspects of neural networks in relation to the proposed representational consistency quantity. I applaud the efforts made by authors.

Improvements/questions:
D) The submission uses a few different phrases in close connection or interchangeably, eg "fuzziness", "order", "linear/nonlinear transformation", "easy/hard to guess/estimation" (see the last 2 paragraphs of page 2). While I understand what the authors are trying to convey, those concepts are not in principle necessarily the same, so some clarification/unification would make the presentation more solid. For instance: nonlinear<--> higher order<-->fuzzy<-->hard to guess. Also guess<-->estimate.

E) Sec 3 is the most important part of the paper and the technical meat. However, I found it harder to follow compared to the rest of the paper. It probably deserves more than 1 page. Authors could consider smoothing the presentation and adding details even at the cost of slightly extending the length.

F) I think section 4.2 could walk the reader through more details to make sure the basics are understood, as this is the first time results of the method are being presented. E.g. fig 3 could be analyzed further and the color maps should be defined.

G) I found the post hoc explanation in the last paragraph of page 6 somewhat dubious.

H) In the feature refinement experiment (paragraph 2 of sec 4.3), is the process done in a recursive manner (first layer 1, then layer 2, and so on)? Or as a one time process? The former seems stronger.

I) The consistency quantity is based on comparing different networks trained for the *same task*. Have you considered doing the same among different networks trained for *different tasks*? Would that say something about similarities among tasks and multi-task learning in a fashion similar to the analysis of Taskonomy 2018?

J) I did not understand "and Beyond" in the title. I'd consider a more directly descriptive title.

K) Seems like authors define "knowledge" and "visual concept" to be the invariant part of a feature (see paragraph 3 of intro). I don't see a particular problem with that, but a direct statement like 'invariant features' would have resonated with me just fine, while whether we can call that "knowledge" is a matter of (unnecessary) semantics in my opinion.


=====
Post rebuttal comments:

Thanks to authors for the detailed response and additions. I read through the comments and skimmed the revised PDF. Overall I preserve my positive rating toward this paper. The updates did improve the paper, but I would recommend the authors to use the camera ready opportunity to further address the original comments especially with respect to the delivery.

I find the added experiments in multi task learning interesting especially given the time constraints of rebuttal. However the adopted setting is not the strongest case as what is defined to be "two tasks" is basically a superclass classification of a fine grained classifier. More concrete datasets and tasks specifically targeting challenging multi task learning are available now (see the full discussion in the original review). I would recommend taking those into consideration either for more experimentation or calibrating the conclusion made out of the current experiment.

I recommend acceptance.

**Experience Assessment:**

I have published in this field for several years.

**Review Assessment: Checking Correctness Of Derivations And Theory:**

I assessed the sensibility of the derivations and theory.

**Review Assessment: Checking Correctness Of Experiments:**

I carefully checked the experiments.

**Review Assessment: Thoroughness In Paper Reading:**

I read the paper thoroughly.

---

> ### Author Response · Authors · 2019-11-12
> **Thank you for your appreciation of our paper. We are glad to answer all your questions.**
>
> Thank you for your appreciation of our paper. We are glad that you considered our method “insightful, well executed, and backed up by many experimental results.” We are glad to answer all your questions.
>
>
> Q: Suggestions about paper writing.
> The first suggestion about paper writing: “The submission uses a few different phrases in close connection or interchangeably, eg ‘fuzziness,’ ‘order,’ ‘linear/nonlinear transformation,’ ‘easy/hard to guess/estimation’. While I understand what the authors are trying to convey, those concepts are not in principle necessarily the same, so some clarification/unification would make the presentation more solid  ...  Also guess<-->estimate.”
> The second suggestion about paper writing: “Sec 3 is the most important part of the paper and the technical meat. However, I found it harder to follow compared to the rest of the paper. It probably deserves more than 1 page. Authors could consider smoothing the presentation and adding details even at the cost of slightly extending the length.”
> The third suggestion about paper writing: “I think section 4.2 could walk the reader through more details to make sure the basics are understood, as this is the first time results of the method are being presented. E.g. fig 3 could be analyzed further and the color maps should be defined.”
> The fourth suggestion about paper writing: “Seems like authors define ‘knowledge’ and ‘visual concept’ to be the invariant part of a feature (see paragraph 3 of intro).”
>
> A: Your suggestions are of great value, and we have followed your suggestions to polish the language. We have corrected above confusing phrases. For example, we have replaced the word “guess” with the word “prediction.” Please see the revised paper for details.
>
> Besides, we have added more explanations about Equation (4) at the end of Section 3. We have added more discussions about consistent feature components and inconsistent feature components in the second paragraph of Section 4.2.
>
> In addition, we have added detailed explanations for results in the caption of Figure 3. We find that blind spots include patterns of the bird torso. In comparison, feature maps of unreliable features mainly consist of random noises.
>
> Finally, for the use of the word “knowledge,” we think “knowledge” is more vivid and meets the people’s cognition.
>
> Q: “I found the post hoc explanation in the last paragraph of page 6 somewhat dubious.”
>
> A: It is an interesting question. Given massive training annotations, deep neural networks with sophisticated architectures usually exhibit better performance. However, when people learn DNNs with limited training samples, the situation is different. As shown in our experiments, shallower neural networks (e.g., the AlexNet) were learned more stably. In comparison, the deep neural network (e.g., the ResNet-34) was more likely to suffer from over-fitting. I.e., due to the significant flexibility of the ResNet-34, ResNet-34 models with different initial parameters were converged to different sub-optimals.
>
>
> Q: “In the feature refinement experiment (paragraph 2 of sec 4.3), is the process done in a recursive manner (first layer 1, then layer 2, and so on)? Or as a one time process? The former seems stronger.”
>
> A: The training process was conducted in a one-time manner. In order to enable clear and rigorous comparisons, we disentangled consistent feature components only from the feature of a single layer as the refined feature. It was because the simultaneous refinement of features of multiple layers would increase the difficulty of clarifying the refinement of which feature made the major contribution. We have introduced above details in the second paragraph of Section 4.3.
>
>
> Q: “The consistency quantity is based on comparing different networks trained for the *same task*. Have you considered doing the same among different networks trained for *different tasks*? Would that say something about similarities among tasks and multi-task learning in a fashion similar to the analysis of Taskonomy 2018?”
>
> A: A good suggestion. We have followed your suggestions and added the multi-task experiment (i.e., the fine-grained classification and the binary classification). In the experiment, we trained DNN A (a VGG-16) to classify 320 species (including 200 bird species and 120 dog species) in a fine-grained manner, and learned DNN B (another VGG-16) for the binary classification of the bird and the dog. Two DNNs were trained on the same datasets but with different labels. We visualized the feature maps of consistent and inconsistent feature components in Figure 5. Please see Section 4.6 for more details about the experiment.
>
>
> Q: “I did not understand ‘and Beyond’ in the title. I'd consider a more directly descriptive title.”
>
> A: Thank you. We have deleted the phrase “and beyond” from the title.

---

### Official Review · AnonReviewer4 · 2019-11-07
**Official Blind Review #4**

**Rating:** 6

**Review:**

This paper presents a method to disentangle intermediate features between two different deep neural networks. More specifically, given two networks, the proposed approach aims to find consistent and inconsistent feature components for a certain layer in each network. If one network is more powerful to the other (e.g., ResNet and AlexNet), the method can figure out which components are weak or strong (i.e., helpful to the performance) for the given task. The authors design a simple yet effective algorithm for extracting knowledge consistency. In addition, they provide a variety of practical experiments including network diagnosis, feature refinement, and network compression. Most of the experimental results support that the proposed method can practically extract consistent feature components with promising improvements.

Main concerns:

1. In section 4.1, the authors provide both the unreliable features and blind spots with their visualization (Figure 3). It is expected that blind spots are more informative than the unreliable features since they are part of a more powerful DNN. However, in Figure 3, the Blind spots do not seem to be important features. This makes the lack of the motivation of visualizing the effectiveness of features. Could you provide more promising visualizing results?

2. In equation (1), it seems that the scaling factor p^(k+1) can be unnecessary because ReLU is scale-invariant and \Sigma_(k+1) can scale the feature h^(k+1) with p_(k+1). Does p_(k+1) significantly improve the performance or negligible?

3. In Table 3, using x^{(0)} to the refined feature shows the best result under VGG-16. Are the result in Table 4 improved when changing the refined features? E.g., only x^{(0)} or x^{(0)} + x^{(1)}?

In short, the proposed disentangle method is simple, novel and very useful in many practical applications as provided in the experimental section. But, some experiments and intuitions of network construction are not fully promising. Thus, I vote for weak acceptance.

Minor concerns:

1. Does the proposed method show meaningful results when the tasks are different but the datasets are the same? Or the same task but different datasets.

2. The disentanglement network can be constructed with other operations, e.g., h^{(k)} = W^{(k)} concat(x , x^{higher}). Did the author try to design the network with other operations?

3. In the fourth paragraph on the first page, the reference of “Wolchover (2017)” need to change “(Wolchover, 2017)”.

4. The proposed method can be applied in other tasks, not with image datasets. Is the algorithm promising for other tasks?

**Experience Assessment:**

I have read many papers in this area.

**Review Assessment: Checking Correctness Of Derivations And Theory:**

I carefully checked the derivations and theory.

**Review Assessment: Checking Correctness Of Experiments:**

I assessed the sensibility of the experiments.

**Review Assessment: Thoroughness In Paper Reading:**

I read the paper thoroughly.

---

> ### Author Response · Authors · 2019-11-12
> **Thank you for your careful review. We will try our best to answer all your questions.**
>
> Thank you for your careful review. The proposed method in our paper is indeed simple, novel, and useful in many applications. We will try our best to answer all your questions.
>
>
> Q: “However, in Figure 3, the Blind spots do not seem to be important features. This makes the lack of the motivation of visualizing the effectiveness of features. Could you provide more promising visualizing results?”
>
> A: A good question. We have followed your suggestions to add more analysis about visualization results in Figure 3. In Figure 3, we can find that blind spots include patterns of the bird’s torso. Whereas, unreliable features usually correspond to noises.
>
> In addition, we have also added a new experiment in Section 4.6, in which we visualized consistent and inconsistent feature components disentangled between two DNNs learned for different tasks.
>
>
> Q: “In equation (1), it seems that the scaling factor p^(k+1) can be unnecessary because ReLU is scale-invariant and \Sigma_(k+1) can scale the feature h^(k+1) with p_(k+1). Does p_(k+1) significantly improve the performance or negligible?”
>
> A: The scaling factor p^(k+1) is crucial in this study. We have clarified the reason in the paragraph under Equation (2). Note that low-order consistent feature components can also be represented by high-order network branches. Therefore, for a rigorous analysis of knowledge consistency, we used p_(k+1) to penalize neural activations from high-order network branches. In this way, we forced as much low-order consistent knowledge as possible to be represented by low-order branches.
>
> Q: “In Table 3, using x^{(0)} to the refined feature shows the best result under VGG-16. Are the result in Table 4 improved when changing the refined features? E.g., only x^{(0)} or x^{(0)} + x^{(1)}?”
>
> A: Thank you. We have followed your suggestions to conduct a new experiment, in which we tested the performance of using x^{(0)} and that of using x^{(0)} + x^{(1)} after removal of redundant features from DNNs. We applied experimental settings for “Mix-CUB” in Table 4, i.e., those for removing dog features from a DNN for the fine-grained classification of both birds and dogs. Please see Section 4.3 for detailed experimental settings. The classification accuracy of “Mix-CUB” w.r.t. the ResNet-18 was 37.63% based on x^{(0)} and 38.28% based on x^{(0)} + x^{(1)}. The classification accuracy of “Mix-CUB” w.r.t. the ResNet-34 was 32.88% based on x^{(0)} and 33.53% based on x^{(0)} + x^{(1)}. The classification accuracy of “Mix-CUB” w.r.t. the VGG-16 conv4-3 was 45.58% based on x^{(0)} and 46.91% based on x^{(0)} + x^{(1)}. The classification accuracy of “Mix-CUB” w.r.t. the VGG-16 conv5-3 was 45.08% based on x^{(0)} and 46.96% based on x^{(0)} + x^{(1)}. We can see that the difference between the accuracy of using different refined features was negligible, compared to the accuracy difference between the refined feature and the unrefined feature.
>
>
> Q: “Does the proposed method show meaningful results when the tasks are different but the datasets are the same? Or the same task but different datasets.”
>
> A: A good suggestion. We have followed your suggestions and added the multi-task experiment (i.e., for the fine-grained classification and the binary classification). We trained DNN A (a VGG-16) to classify 200 bird species and 120 dog species in a fine-grained manner, and learned DNN B (another VGG-16) for the binary classification of the bird and the dog. We visualized the feature maps of consistent and inconsistent feature components in Figure 5. Please see Section 4.6 for more details of the experiment.
>
>
> Q: “The disentanglement network can be constructed with other operations, e.g., h^{(k)} = W^{(k)} concat(x ,x^{higher}). Did the author try to design the network with other operations?”
>
> A: The replacement of h^{(k)} = W^{(k)} (x + x^{higher}) with h^{(k)} = W^{(k)}concat(x, x^{higher}) will lead to two problems. First, a rigorous analysis of knowledge consistency requires us to force as much low-order consistent knowledge as possible to be represented by low-order branches. It is because low-order consistent feature components can also be represented by high-order branches. However, the replacement makes it difficult to penalize p_(k+1) to push neural activations towards low-order branches.
>
> Second, an distinct contribution is that our method can divide consistent features into feature components of different orders (g(x)= x^{(0)}+ x^{(1)} +x^{(2)}) for further analysis. However, h^{(k)} = W^{(k)}concat(x , x^{higher}) does not support the disentanglement of feature components.
>
>
> Q: “The proposed method can be applied in other tasks, not with image datasets. Is the algorithm promising for other tasks?”
>
> A: Although this paper only focuses on issues in computer vision, theoretically, our method can be applied to DNNs oriented to other types of data. We will try to use our method to diagnose various DNNs in future research. Thank you for your suggestion.

---

### Decision · Program_Chairs · 2019-12-19

**Decision:**

Accept (Poster)

**Comment:**

This paper presents a method for extracting "knowledge consistency" between neural networks and  understanding their representations.

Reviewers and AC are positive on the paper, in terms of insightful findings and practical usages, and also gave constructive suggestions to improve the paper. In particular, I think the paper can gain much attention for ICLR audience.

Hence, I recommend acceptance.